# Validation of the shortened version of the Canine Behavioral Assessment and Research Questionnaire (C-BARQ) using participants from the Dog Aging Project

**Vanessa Wilkins**[1°]**, Jeremy Evans**[1°]**, Christina Park**[2]**, The Dog Aging Project Consortium**[¶]**, Annette L. Fitzpatrick**[2,3]**, Kate E. Creevy**[1]**, Audrey Ruple**[4]*

**1** Department of Small Animal Clinical Sciences, Texas A&M University College of Veterinary Medicine & Biomedical Sciences, College Station, Texas, United States of America, **2** Department of Epidemiology, University of Washington, Seattle, WA, United States of America, **3** Department of Family Medicine, University of Washington, Seattle, WA, United States of America, **4** Department of Population Health Sciences, Virginia-Maryland College of Veterinary Medicine, Virginia Polytechnic Institute and State University, Blacksburg, Virginia, United States of America

☯ These authors contributed equally to this work.
¶ A list of the Dog Aging Project Consortium members and their affiliations is provided in the Acknowledgments
* aruple@vt.edu

**Data Availability Statement:** All DAP data is anonymized and made available through our open

## Abstract

The Canine Behavioral Assessment and Research Questionnaire (C-BARQ) is a 100-item owner-completed survey instrument used for assessing behavior and temperament of companion dogs. The shortened version of the C-BARQ (C-BARQ[(S)]) consists of 42 items of the long C-BARQ. We aimed to validate the shortened C-BARQ[(S)] by comparing it with the long questionnaire in the same human-dog pair. We examined data from a nationwide cohort of companion dogs enrolled in the large-scale longitudinal Dog Aging Project (DAP) study. Among 435 participating owners who completed both the long and shortened versions of the C-BARQ within 60 days of each other, agreement between individual questions of the long and shortened C-BARQ using an unweighted kappa statistic and percent agreement was examined. Associations between the two questionnaires for mean behavior and temperament domain scores and mean miscellaneous category scores were assessed using Pearson correlation coefficients. Of 435 dogs in the study, the mean (SD) age was 7.3 (4.3) years and 216 (50%) were female. Kappa values between the long and shortened C-BARQ for individual questions within the 14 behavior and temperament domains and a miscellaneous category ranged from fair to moderate (0.23 to 0.40 for 21 items and 0.41 to 0.58 for 26 items, respectively). Pearson correlation coefficients above 0.60 between both questionnaires for 12 of the 14 mean behavior and temperament domain scores and a category of miscellaneous items were observed. Kappa values for individual questions between the long and shortened C-BARQ ranged from fair to moderate and correlations between mean domain scores ranged from moderate to strong.

data access portal which can be found here: https://dogagingproject.org/open_data_access/. We have also included all matched C-BARQ responses for both the short and long versions of the survey with the supplemental materials.

**Funding:** The author(s) received no specific funding for this work.

**Competing interests:** The authors have declared that no competing interests exist.

## Introduction

Behavioral profiles of companion animals are often determined via direct observation by researchers, trainers, veterinarians and pet owners [1,2]. However, direct observation of animals like dogs can be complicated by various factors including the presence of a human observer [3] or restricted environments such as a veterinary setting [4] or laboratory kennels [5], which can elicit unintended responses from animals under observation [6]. Because of these factors, questionnaires, such as the frequently used Canine Behavioral Assessment and Research Questionnaire (C-BARQ), that ask dog owners about their dog's responses to everyday stimuli or situations are increasingly used to obtain information about the behavior and temperament of pet dogs [7].

The C-BARQ was first developed in 2003 by Hsu and Serpell (http://www.cbarq.org) [7] and the current version is a 100-item questionnaire (henceforth referred to as long) that investigates dog temperament and behavior within 14 different behavior and temperament domains and an additional set of miscellaneous behaviors [7,8]. The C-BARQ has been used to examine how canine behavior is associated with diverse variables including early life experiences, manner of acquisition (e.g., from breeders or pet stores), and breed [9–11]. Additionally, the C-BARQ has been shown in prior studies to be a useful tool in identifying behavioral issues associated with relinquishment to shelters [12], predicting success as a service or guide dog [8], and investigating risk factors associated with fear during veterinary visits [13]. The C-BARQ has been translated into multiple languages and utilized successfully in multiple countries, including Japan [14], Mexico [15], Iran [16] and the United States (US) [17].

While owner-completed questionnaires provide researchers the opportunity to study a large sample size of dogs in their usual environment without causing undue stress [17], longer questionnaires may especially burden respondents in potentially distressing situations [18] such as pet relinquishment to an animal shelter [19] and death of a pet [20]. In a study that evaluated the 103-item version of the C-BARQ for behavioral assessment of dogs relinquished to shelters, Segurson *et al.* acknowledged the inconvenience that a questionnaire completion may pose to people at the time of relinquishing their dog [12]. A shortened version of the C-BARQ as a tool to screen for behavioral problems in shelter-relinquished dogs was developed by Duffy *et al.* [21].

Two previous studies have reported the strength of the shortened C-BARQ[(S)] in being able to identify behavioral problems associated with intensive breeding (i.e., puppy farming) as well as dog relinquishment to shelters [21,22]. However, the validity of the C-BARQ[(S)] has yet to be investigated via the comparison of both the long and shortened versions in the same human-dog pair. The purpose of this study was to validate the C-BARQ[(S)] by comparing its results to those obtained from the long C-BARQ in a US-wide cohort of companion dogs participating in the Dog Aging Project (DAP) [23].

## Materials and methods

### Study population

The design and methods of the DAP have been published previously [23]. Briefly, the DAP is a longitudinal study on healthy aging in dogs through the collection of data from owners of enrolled dogs and biospecimens in a subset of dogs. Dogs enrolled in the DAP are collectively called the DAP Pack. In the summer of 2020, new participants were regularly joining the DAP by completing the Health and Life Experience Survey (HLES), which includes the shortened C-BARQ[(S)]. On July 31st, 2020, we invited all DAP participants who joined the project between July 1 and July 30, 2020 to complete the long C-BARQ at the C-BARQ website (https://vetapps.vet.upenn.edu/cbarq/) by creating an account at the C-BARQ site, and using a

unique identification code provided in the email invitation when responding to the survey. Invitations were sent to 749 people who met these criteria. They were sent two reminders over the following week. C-BARQ responses were accepted through August 31, 2020. Responses from participants who had completed both HLES and long C-BARQ within 60 days of each other were included in the study. The present study included 435 study participants who completed both the long and shortened versions of the C-BARQ (response rate of 58.1%).

This work was reviewed by the Texas A&M University Institutional Animal Care and Use Committee and the Animal Use Protocol was approved. The study was reviewed by the University of Washington Institutional Review Board and was exempt from obtaining informed consent from dog owners because the limited information collected in the HLES was human subjects research that met the qualifications for Exempt Status (Category 2).

## Demographics

Information on dog demographics was obtained from the owner-completed baseline HLES. Dog age was measured continuously in years. Sex was defined as a dichotomous variable (male or female). Dog weight was reported as a continuous measure and converted from pounds to kilograms (kg) for analysis. Dog size was categorized according to different weight cutoff values for adult dogs and puppies (assessed using expected adult weight) as follows: Toy and Small (<10 kg), Medium (10 to <20 kg), Standard (20 to <30 kg), Large (30 to <40 kg) or Giant ($\geq$40 kg). Type of breed and sterilization status were defined as dichotomous variables (purebred or mixed and desexed or intact, respectively).

## Long and shortened C-BARQ

Details about the development of the long and shortened versions of the C-BARQ have been previously described [7,8,21]. The long C-BARQ consists of 100 questions divided into 14 behavior and temperament domains (also referred to as subscales or factors) identified by factor analysis and a group of miscellaneous questions. Respondents were asked to rate their dog's responses to specific behaviors in the recent past on a 5-point Likert scale (0 to 4) measuring either severity (0 = no signs to 4 = severe signs of the behavior) or frequency (0 = never to 4 = always). Higher scores reflect less desirable behavior, except for questions related to trainability. Scores for a question that asked owners to rate how trainable and obedient their dog was when "easily distracted by interesting sights, sounds or smells" (higher scores represent greater trainability) were reverse coded so that higher values indicated a less trainable dog. In the long C-BARQ, all questions had an additional response option of "not observed/ not applicable" for the specified situation.

The C-BARQ[(S)] is a shortened 42-item version of the long C-BARQ with a typical completion time of less than 10 minutes [21,22]. The C-BARQ[(S)] was developed using an iterative approach whereby select questions within the long C-BARQ behavior and temperament domains were removed in a stepwise manner [21]. Duffy *et al.* calculated the internal consistency, as measured by Cronbach's alpha, of the reduced questions within the domains with the removal of each of the remaining items [21]. Thirty-five questions in the C-BARQ[(S)] were subscaled into the same 14 behavior and temperament domains in the long C-BARQ: stranger-directed aggression, owner-directed aggression, stranger-directed fear, nonsocial fear, dog-directed aggression, dog-directed fear, dog rivalry, separation-related behavior, attachment and attention-seeking behavior, trainability, chasing, excitability, touch-sensitivity and energy level. The remaining seven questions that were not included in the 14 behavior and temperament domains were grouped into a "Miscellaneous" category.

Five individual questions in the C-BARQ(S) were generated by combining two separate questions in the long questionnaire into a single question. Examples that follow are five individual questions in the C-BARQ(S) with modified words in italics and original words in italics and parentheses: [1] "Aggressive when approached directly by an unfamiliar *person* while being walked/exercised on a leash" (one question for *adult* and one for *child* in the C-BARQ); [2] "Aggressive when approached directly by an unfamiliar dog while being walked/exercised on a leash" (one question for *male* and one for *female* identifiers placed before "dog" in the C-BARQ); [3] "Afraid or anxious when approached directly by an unfamiliar *person* while away from your home" (one question for *adult* and one for *child* in the C-BARQ); [4] "Afraid or anxious when approached directly by an unfamiliar dog" (one question for *same or larger size* and one for *smaller size* identifiers placed before "dog" in the C-BARQ); and [5] "*Barking/whining* when left alone for any period of time" (one question for *barking* and one for *whining* in the C-BARQ).

## Statistical analyses

Characteristics of dogs were described as mean (standard deviations [SD]) or as frequency (percentages), where appropriate.

Individual questions in both the long and shortened versions of the C-BARQ were directly compared to each other to evaluate if the participants agreed with themselves, when asked the same question in two different questionnaires. For single questions in the C-BARQ(S) that were combinations of separate questions in the long C-BARQ, they were compared to each of the separate questions in the C-BARQ. Intraobserver agreement on ordinal-scale responses to individual questions in both questionnaires was assessed using the unweighted kappa statistic and percent agreement. It is important to note that while the kappa statistic has been commonly referred to as a measure of agreement, it is a measure of the amount by which the observed agreement deviates from the agreement expected by chance [24]. In contrast, the percent agreement represents the observed agreement that does not account for concordance produced by chance alone [25]. To help put the kappa into perspective, we present the kappa with the percent agreement [26].

The kappa statistic can range from -1 to 1 (usually 0 to 1) and was interpreted according to commonly cited labels suggested by Landis and Koch [27]: poor ($<0$), slight (0.00–0.20), fair (0.21–0.40), moderate (0.41–0.60), substantial (0.61–0.80) and almost perfect (0.81–1.00). Percent agreement between the long and shortened C-BARQ for individual questions was calculated as the number of participants who reported the same score in both versions of the survey divided by the total number of participants who provided a response to the question, multiplied by 100.

Mean domain scores corresponding to each behavior and temperament domain were calculated by summing the domain-specific responses and dividing by the number of questions in the domain and presented with SD. The number of questions included in mean domain-specific scores differed between the long and shortened C-BARQ given the varying lengths of the surveys. All questions in the long form were used in the calculation of the mean domain score regardless of the number of items in the equivalent domain of the short form.

The direction and strength of correlations between mean domain scores were quantified using Pearson correlation coefficients. Pearson correlation values were classified as very weak (0–0.19), weak (0.20–0.39), moderate (0.40–0.59), strong (0.60–0.79) and very strong (0.80–1) [28]. Scatterplots with a fitted regression line were used to visualize the associations of mean domain scores between C-BARQ and C-BARQ(S) for correlation coefficients that were the largest (excluding the miscellaneous category) and smallest in our data.

We found that not all dog owners responded to every question in both the long and shortened versions of the C-BARQ. As a result, only the questions that were answered in both

versions were included in the analysis. Additionally, "not observed/not applicable" responses were treated as missing. We considered a two-tailed p-value of <0.05 to be statistically significant. All analyses were conducted using IBM SPSS Statistics for Windows, Version 19 (IBM Corp., Armonk, NY) and R statistical software version 4.1.1 (R Foundation for Statistical Computing, Vienna, Austria).

### Ethics

This study utilizes data obtained from dog owners about their pet dogs. Written consent for this work was obtained and no human level data was collected. This work was reviewed by the Texas A&M University Institutional Animal Care and Use Committee and the Animal Use Protocol (IACUC 2021–0317 CA) was approved. The study was reviewed by the University of Washington Institutional Review Board and was exempt from obtaining informed consent from dog owners because the limited information collected in the HLES was human subjects research that met the qualifications for Exempt Status (Category 2).

### Results

A total of 435 DAP dog owners who completed both the long and shortened versions of the C-BARQ were included in this study. Characteristics of the dogs stratified by sex are presented in Table 1. The mean (SD) age of the dogs was 7.3 (4.3) years, with almost equal distribution of sex. Compared to male dogs, female dogs tended to weigh less; were less likely to be large or giant sized and more likely to be medium sized. The majority of owners were aged between 45 and 74 at the time they completed the survey, had at least a Bachelor's degree, and earned at least $60,000 per year (Table 1).

Kappa values between individual questions across all 14 behavior and temperament domains plus the miscellaneous items in both questionnaires varied from fair to moderate (Table 2). There were 21 questions with kappa values that were considered fair (kappa ranged from 0.23 to 0.40) and 26 questions with kappa values that were moderate (kappa ranged from 0.41 to 0.58). The lowest kappa values between individual questions in both questionnaires were seen for questions in the domain of excitability: "becomes excitable just before being taken for a walk" (kappa = 0.25) and "becomes excitable just before taken for a car ride" (kappa = 0.23).

Regarding the question "Aggressive when approached directly by an unfamiliar *person* while being walked/exercised on a leash," which was asked as a single question (*person*) in the shortened C-BARQ[(S)] or as separate questions (*adult* or *child*) in the long C-BARQ, the kappa value between questions about *person* and *adult* (kappa = 0.55) was higher than the kappa value between *person* and *child* (kappa = 0.36). On the other hand, regarding the question "Afraid or anxious when approached directly by an unfamiliar *person* while away from your home," the kappa value between questions about *person* and *adult* (kappa = 0.46) was similar to the kappa value between *person* and *child* (kappa = 0.43).

Percent agreement between questions in C-BARQ and C-BARQ[(S)] was high (>80%) for kappa values in the range of 0.35 to 0.58 for nine questions in the domains of stranger-directed aggression, owner-directed aggression and separation-related behavior and the miscellaneous category (urination, defecation and tail chasing) (Table 2). For all nine of these questions in C-BARQ and C-BARQ[(S)], the majority of responses fell into one category of no signs/never.

Pearson correlation coefficients for mean domain scores between both questionnaires were above 0.60 (range: 0.65 to 0.83) for 12 behavior and temperament domains that will follow and a miscellaneous category: stranger-directed aggression, stranger-directed fear, nonsocial fear, dog-directed aggression, dog-directed fear, dog rivalry, separation-related behavior,

**Table 1. Characteristics by sex in 435 dogs.**

| Characteristic | Female (N = 216) | Male (N = 219) | Total (N = 435) |
|---|---|---|---|
| Age (years), mean (SD) | 7.4 (4.3) | 7.1 (4.3) | 7.3 (4.3) |
| Weight (kilograms), mean (SD) | 20.9 (11.1) | 24.3 (14.3) | 22.6 (12.9) |
| Size[a], n (%) | | | |
| Toy and small | 43 (19.9) | 54 (24.7) | 97 (22.3) |
| Medium | 48 (22.2) | 28 (12.8) | 76 (17.5) |
| Standard | 66 (30.6) | 67 (30.6) | 133 (30.6) |
| Large | 51 (23.6) | 57 (26.0) | 108 (24.8) |
| Giant | 8 (3.7) | 13 (5.9) | 21 (4.8) |
| Type of breed, n (%) | | | |
| Purebred | 105 (48.8) | 110 (50.2) | 215 (49.4) |
| Mixed | 111 (51.4) | 109 (49.8) | 220 (50.6) |
| Sterilization status, n (%) | | | |
| Desexed | 203 (94.0) | 201 (91.8) | 404 (92.9) |
| Intact | 13 (6.0) | 18 (8.2) | 31 (7.1) |
| Owner age in years, n (%) | | | |
| 18–24 | 0 (0.0) | 3 (1.4) | 3 (0.7) |
| 25–34 | 23 (10.7) | 18 (8.2) | 41 (9.4) |
| 35–44 | 33 (15.2) | 47 (21.5) | 80 (18.4) |
| 45–54 | 43 (19.9) | 47 (21.5) | 90 (20.7) |
| 55–64 | 56 (25.9) | 55 (25.1) | 111 (25.5) |
| 65–74 | 54 (25.0) | 42 (19.2) | 96 (22.1) |
| 75 and older | 7 (3.2) | 7 (3.2) | 14 (3.2) |
| Owner highest education level, n (%) | | | |
| No college degree | 28 (13.0) | 18 (8.2) | 46 (10.6) |
| Associate degree or trade training | 15 (6.9) | 22 (10.1) | 37 (8.5) |
| Bachelor's degree | 54 (25.0) | 74 (33.8) | 128 (29.4) |
| Master's degree | 71 (32.9) | 75 (34.2) | 146 (33.6) |
| Professional degree (DVM, MD, DDS, or JD) | 25 (11.6) | 16 (7.3) | 41 (9.4) |
| Doctorate degree (PhD, DrPH, or DPhil) | 23 (10.7) | 14 (6.4 | 37 (8.5) |
| Owner annual income, n (%) | | | |
| Less than $59,999 | 31 (14.4) | 33 (15.1) | 64 (14.7) |
| $60,000 - $119,999 | 71 (32.9) | 80 (36.5) | 151 (34.7) |
| $120,000 - $179,999 | 44 (20.4) | 44 (20.1) | 88 (20.2) |
| $180,000 or more | 41 (19.0) | 40 (18.3) | 81 (18.6) |
| Preferred not to answer | 29 (13.4) | 22 (10.1) | 51 (11.7) |

a Dog size was defined based on adult weight cutoff values: Toy and Small (<10 kg), Medium (10 to <20 kg), Standard (20 to <30 kg), Large (30 to <40 kg) or Giant (≥40 kg).

attachment and attention-seeking behavior, trainability, chasing, touch sensitivity and energy level (Table 3). In addition, mean domain scores for owner-directed aggression and excitability were moderately correlated between the two questionnaires (Pearson correlation coefficients = 0.52 and 0.57, respectively).

Fig 1A and 1B depict scatterplots of mean domain scores in C-BARQ[(S)] (x-axis) versus C-BARQ (y-axis) for owner-directed aggression (Pearson correlation coefficient = 0.52) and stranger-directed fear (Pearson correlation coefficient = 0.81). Mean domain scores for

**Table 2. Kappa statistic and percent agreement for individual questions in the long C-BARQ and shortened C-BARQ(S) among participants of the Dog Aging Project (DAP).**

| Question | N | Kappa | Percent Agreement |
|---|---|---|---|
| **Domain 1. Stranger-directed aggression** | | | |
| Aggressive when approached directly by an unfamiliar *person* (vs. *adult* in C-BARQ) while being walked/exercised on a leash* | 425 | 0.55 | 80.0 |
| Aggressive when approached directly by an unfamiliar *person* (vs. *child* in C-BARQ) while being walked/exercised on a leash* | 412 | 0.36 | 74.3 |
| Aggressive when mailmen or other delivery workers approach your home | 425 | 0.50 | 65.4 |
| Aggressive when strangers walk past your home while your dog is outside or in the yard | 406 | 0.44 | 62.3 |
| **Domain 2. Owner-directed aggression** | | | |
| Aggressive when toys, bones or other objects are taken away by a household member | 432 | 0.45 | 86.6 |
| Aggressive when approached directly by a household member while s/he (the dog) is eating | 429 | 0.47 | 95.3 |
| Aggressive when his/her food is taken away by a household member | 420 | 0.48 | 93.8 |
| **Domain 3. Stranger-directed fear** | | | |
| Afraid or anxious when approached directly by an unfamiliar *person* (vs. *adult* in C-BARQ) while away from your home* | 424 | 0.46 | 71.2 |
| Afraid or anxious when approached directly by an unfamiliar *person* (vs. *child* in C-BARQ) while away from your home* | 409 | 0.43 | 71.1 |
| Afraid or anxious when an unfamiliar person tries to touch or pet your dog | 432 | 0.47 | 71.5 |
| **Domain 4. Nonsocial fear** | | | |
| Afraid or anxious in response to sudden or loud noises (e.g. vacuum cleaner, car backfire, road drills, objects being dropped, etc.) | 431 | 0.32 | 47.3 |
| Afraid or anxious in response to strange or unfamiliar objects on or near the sidewalk | 422 | 0.42 | 67.3 |
| Afraid or anxious when first exposed to unfamiliar situations (e.g. first car trip, first time in elevator, first visit to veterinarian, etc.) | 425 | 0.33 | 53.4 |
| **Domain 5. Dog-directed aggression** | | | |
| Aggressive when approached directly by an unfamiliar (vs. *male* in C-BARQ) dog while being walked/exercised on a leash* | 414 | 0.45 | 61.8 |
| Aggressive when approached directly by an unfamiliar (vs. *female* in C-BARQ) dog while being walked/exercised on a leash* | 415 | 0.45 | 62.2 |
| Aggressive when barked, growled, or lunged at by another (unfamiliar) dog | 403 | 0.39 | 52.1 |
| **Domain 6. Dog-directed fear** | | | |
| Afraid or anxious when approached directly by an unfamiliar dog (vs. of the *same or larger* size in C-BARQ)* | 420 | 0.34 | 56.7 |
| Afraid or anxious when approached directly by an unfamiliar dog (vs. of a *smaller* size in C-BARQ)* | 415 | 0.31 | 56.9 |
| Afraid or anxious when barked, growled, or lunged at by an unfamiliar dog | 393 | 0.30 | 46.3 |
| **Domain 7. Dog rivalry** | | | |
| Aggressive when approached while eating by another (familiar) household dog | 246 | 0.44 | 72.8 |
| Aggressive when approached while playing with/chewing a favorite toy, bone, object, etc., by another (familiar) household dog | 254 | 0.51 | 74.0 |
| **Domain 8. Separation-related behavior** | | | |
| Restlessness, agitation or pacing when left alone for any period of time | 422 | 0.33 | 63.5 |
| Barking/whining (vs. *whining* in C-BARQ) when left alone for any period of time* | 424 | 0.33 | 57.5 |
| Barking/whining (vs. *barking* in C-BARQ) when left alone for any period of time* | 425 | 0.33 | 57.6 |
| Chewing or scratching at doors, floors, windows, curtains, etc. when left alone for any period of time | 427 | 0.35 | 80.6 |
| **Domain 9. Attachment and attention-seeking behavior** | | | |

*(Continued)*

**Table 2.** (Continued)

| Question | N | Kappa | Percent Agreement |
|---|---|---|---|
| Tends to sit close to, or in contact with, you (or others) when you are sitting down | 433 | 0.34 | 53.8 |
| Tends to follow you (or other members of household) about the house, from room to room | 430 | 0.41 | 56.3 |
| **Domain 10. Trainability** | | | |
| Easily distracted by sights, sounds, or smells[b] | 433 | 0.30 | 50.1 |
| Obeys the "sit" command immediately | 427 | 0.47 | 66.3 |
| Obeys the "stay" command immediately | 418 | 0.42 | 58.4 |
| **Domain 11. Chasing** | | | |
| Chases or would chase birds given the opportunity | 426 | 0.35 | 48.4 |
| Chases or would chase squirrels, rabbits, and other small animals given the opportunity | 425 | 0.46 | 60.7 |
| Escapes or would escape from home or yard given the chance | 399 | 0.53 | 65.9 |
| **Domain 12. Excitability** | | | |
| Becomes excitable just before being taken for a walk | 423 | 0.25 | 42.6 |
| Becomes excitable just before being taken on a car trip | 429 | 0.23 | 41.0 |
| **Domain 13. Touch sensitivity** | | | |
| Afraid or anxious when having nails clipped by a household member | 344 | 0.45 | 59.3 |
| Afraid or anxious when groomed or bathed by a household member | 415 | 0.40 | 63.6 |
| **Domain 14. Energy level** | | | |
| Hyperactive, restless, has trouble settling down | 431 | 0.38 | 66.1 |
| Playful, puppyish, boisterous | 433 | 0.36 | 54.0 |
| Active, energetic, always on the go | 427 | 0.37 | 53.4 |
| **Miscellaneous** | | | |
| Chews inappropriate objects | 430 | 0.41 | 65.3 |
| Pulls excessively hard when on the leash | 429 | 0.43 | 58.5 |
| Urinates against objects/furnishings in your home | 428 | 0.54 | 92.1 |
| Urinates when left alone at night, or during the daytime | 429 | 0.58 | 89.3 |
| Defecates when left alone at night, or during the daytime | 425 | 0.58 | 90.6 |
| Chases own tail/hind end | 421 | 0.56 | 86.0 |
| Barks persistently when alarmed or excited | 433 | 0.37 | 51.3 |

*Each of the five individual questions in the shortened C-BARQ[(S)] was derived from two separate questions in the long C-BARQ (original words are in italics and in parentheses). The kappa statistic and percent agreement were calculated between each of the five questions in the C-BARQ[(S)] and each of the related single questions in the long C-BARQ.

[a]The trainability question, "Easily distracted by sights, sounds, or smells", was reverse-scored so that higher scores reflect poorer trainability.

P-values of kappa statistics for all questions were <0.001.

owner-directed aggression in C-BARQ and C-BARQ[(S)] showed low variability (many zeros) (**Fig 1A**). Compared with owner-directed aggression, greater variability in the C-BARQ and C-BARQ[(S)] mean domain scores for stranger-directed fear and a stronger linear trend was observed, although there was still a considerable number of zeros (**Fig 1B**).

## Discussion

The present study aimed to validate the shortened version of the C-BARQ by administering the shortened and long versions of the questionnaire to owners of companion dogs enrolled in

**Table 3. Pearson correlation coefficients of mean domain scores between the shortened C-BARQ(S) and long C-BARQ among participants of the Dog Aging Project (DAP).**

| Domain | N | C-BARQ(S) Mean[a] (SD) | C-BARQ Mean[a] (SD) | Pearson Correlation Coefficient |
|---|---|---|---|---|
| Stranger-directed aggression | 346 | 0.77 (0.74) | 0.51(0.54) | 0.77 |
| Owner-directed aggression | 279 | 0.10 (0.27) | 0.15 (0.21) | 0.52 |
| Stranger-directed fear | 407 | 0.62 (0.91) | 0.48 (0.76) | 0.81 |
| Nonsocial fear | 363 | 1.05 (0.79) | 0.89 (0.68) | 0.74 |
| Dog-directed aggression | 254 | 1.28 (1.06) | 0.86 (0.77) | 0.75 |
| Dog-directed fear | 326 | 1.13 (1.02) | 0.83 (0.90) | 0.70 |
| Dog rivalry | 240 | 0.55 (0.77) | 0.44 (0.70) | 0.73 |
| Separation-related behavior | 392 | 0.59 (0.76) | 0.42 (0.52) | 0.68 |
| Attachment and attention-seeking behavior | 380 | 2.71 (0.92) | 1.96 (0.70) | 0.65 |
| Trainability[b] | 370 | 2.36 (0.77) | 2.62 (0.56) | 0.68 |
| Chasing | 387 | 1.89 (1.01) | 1.92 (0.99) | 0.79 |
| Excitability | 383 | 2.25 (0.92) | 2.01 (0.79) | 0.57 |
| Touch sensitivity | 339 | 1.02 (1.02) | 0.69 (0.75) | 0.69 |
| Energy level | 425 | 1.39 (0.80) | 1.43 (0.76) | 0.74 |
| Miscellaneous[c] | 406 | 0.61 (0.40) | 0.61 (0.43) | 0.83 |

[a]The number of individual questions included in the calculation of mean domain scores differed between the shortened and long C-BARQ given the varying lengths of the surveys.

[b]The trainability mean domain score includes a reverse-coded item ("Easily distracted by sights, sounds, or smells") so that high scores indicate less trainable.

[c]The Miscellaneous category contains seven behaviors: Chews inappropriate objects; pulls excessively hard when on the leash; urinates against objects/furnishings in your home; urinates when left alone at night, or during the daytime; defecates when left alone at night, or during the daytime; chases own tail/hind end; and barks persistently when alarmed or excited.

P-values for all correlation coefficients were <0.001.

the national DAP. We found that kappa values for individual questions about behavior asked in the shortened and long C-BARQ ranged from fair to moderate and correlation coefficients

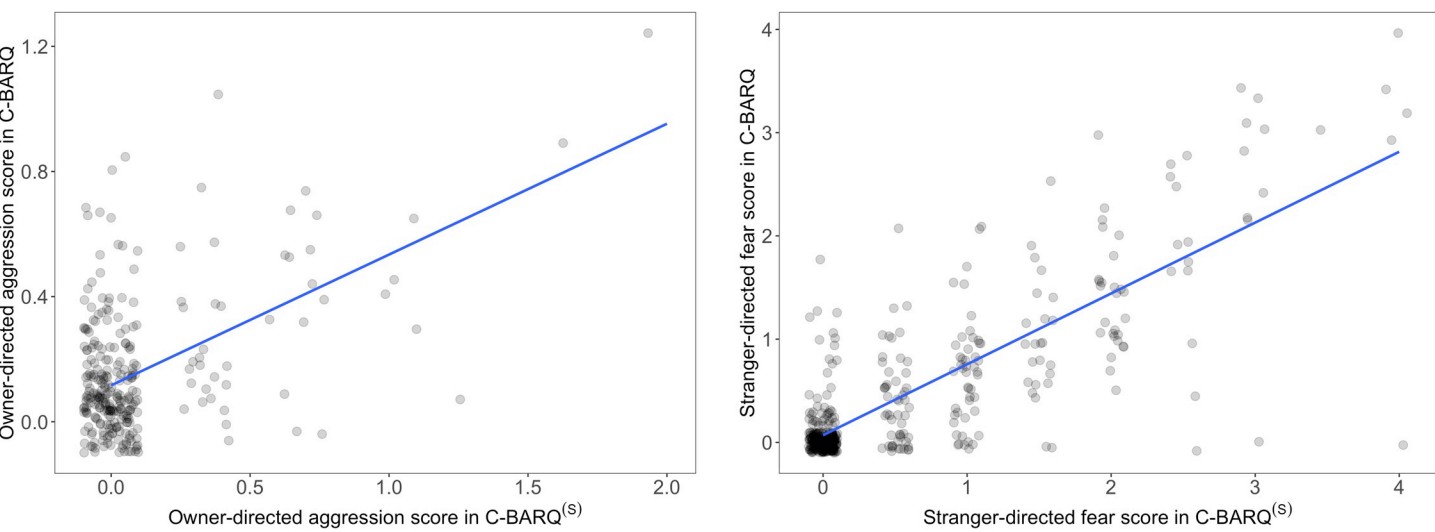

**Fig 1.** Scatterplots of mean domain scores between C-BARQ(S) (x-axis) and C-BARQ (y-axis) for (A) owner-directed aggression and (B) stranger-directed fear. (Notes: For visual purposes only, a small amount of random variation (jitter) was added to each point to make the points easier to read. The x-axis and y-axis scales are not the same in 1A and 1B).

were greater than 0.60 for nearly all mean domain scores plus mean miscellaneous category scores in both versions of the C-BARQ. These results support that the shortened C-BARQ[(S)] has comparable validity to the long version of the C-BARQ.

To our knowledge, our study is the first to conduct an external validation of the shortened C-BARQ[(S)] by comparing scores between the same dog-owner pair for the long and shortened versions of the C-BARQ in an independent sample. Two prior studies demonstrated the validity of the shortened C-BARQ[(S)] using an existing dataset [21] and samples of shelter dogs [21] and puppy farm dogs [22]. Duffy *et al.* found strong correlations between the long and shortened C-BARQ domains scores calculated using data obtained from a convenience sample of currently-owned dogs (N = 17,307) in the C-BARQ database [21]. In the same study but using a different sample of 438 dogs relinquished to three shelters, Duffy *et al.* reported adequate-good internal reliability (Cronbach's alpha), in all but one of the 14 domains of the C-BARQ[(S)] [21]. In another study of 2,026 dogs from different backgrounds (i.e., non-puppy farms, puppy farms and unknown), Wauthier *et al.* conducted a replication study of Duffy *et al.*'s shortened C-BARQ[(S)] [21] and showed that their extracted domains and domain reliability calculated using Cronbach's alpha were consistent with results previously reported by Duffy *et al.* [22].

To date, there exists a number of brief questionnaires about personality and behavior of companion dogs as reported by owners in the literature. In addition to the shortened C-BARQ[(S)], the two other frequently used short assessments of canine personality and behavioral surveys include the 26-item Monash Canine Personality Questionnaire-revised (MCPQ-R) [29] and the 45-item Dog Personality Questionnaire (DPQ), both of which have undergone multiple reductions in the number of questions from their original lengths of 67 (to 41 and then 26) and 102 (to 75 and then 45) items [30]. Based on knowledge from research involving human subjects, questionnaires that were short and easy to complete were generally associated with higher completion rates [31] and lower response burden while the shorter length did not compromise the validity and reliability of the questionnaire [32]. However, while inter-rater and test-retest reliability and validity have been shown in both the MCPQ-R [29,33] and DPQ [30], including strong convergence between analogous personality constructs of the MCPQ-R and DPQ [34], both questionnaires focus on the assessment of canine personality [35,36] according to personality dimensions that were adapted from the Five-Factor Model in human personality research [37,38]. In contrast, the shortened C-BARQ[(S)] does not only restrict itself to domains of temperament and personality of the dog but also works as an alternative to behavioral testing and direct assessment of dog behavior by leveraging owners' observations of their dog's behavioral response to specific situations.

Strengths of our study include its comprehensive assessment of a wide range of canine behaviors in the same dog-owner pair of a nationwide sample; examination of a wide range of dog age and breed type; and adequate response rate.

There are several limitations of this study that should be acknowledged. First, given that our study occurred in the summer of 2020 during the extremely variable COVID-19 lockdown policies, it is possible that the timing and type of the lockdown measures may have affected dog behavior [39] as a result of change in dog management practices including walking duration and frequency and dog-dog and dog-human interactions [40]. Hence, our findings should be considered in the context of a drastically changed environment due to the COVID-19 pandemic. Second, while the response rate at nearly 60% was acceptable, particularly given the timing of the survey request in relation to world events, the potential for non-response bias is worth mentioning. For instance, it could be that participants that were not able to complete the long version of the C-BARQ survey were those who were no longer under quarantine restrictions in place due to the pandemic and thus their dogs may have been experiencing a shift in their daily routine that could have led to less agreement between survey responses than

within the population who was able to complete both surveys. Third, as the DAP Pack is a self-selected group of dogs and dog owners who voluntarily responded to calls for participants, findings from the study may have limited generalizability to the broader population of companion dogs and their owners living in the US. Fourth, given that surveys about behavior and personality traits are susceptible to social desirability bias, there may have been some underreporting of aggressive behaviors observed in dogs by owners. However, as this was a web-based self-administered questionnaire, the potential of social desirability bias was likely reduced [41]. Fifth, the size of the correlation coefficient is subject to various factors, one of which is the amount of variability in the two sets of measurements whose association is being evaluated [42]. As many of the owners' ratings of their dog's response to specific situations dominated one category, there was a large number of zeros in our data, and thus may provide one explanation as to why we did not see higher correlation coefficients. Sixth, the kappa statistic has been criticized for its susceptibility to two paradoxes [43,44]. Kappa can be low when observed agreement is high or kappa can be high when there is imbalance in the marginal totals of a contingency table (i.e., 2x2 table) [43]. Reasons include the dependency of the kappa statistic on the prevalence of the characteristic under study [45] or the distribution of marginal totals in the table [46]. Our observed kappa values reflect the first paradox of low kappa values despite high observed percent agreement. Agreement due to chance alone in our study was high, and thus agreement over and above chance was unlikely. As seen in our study, most of the dog owners' repeated ratings of their dog's behavior fell into the same category. It is possible that the frequency and severity of behaviors assessed were rarely observed in participating dogs by their owners. Kappa values may have also depended on the marginal distributions, a direct consequence of the definition of kappa [46]. Therefore, as recommended in the literature [26], we have presented both the kappa and percent agreement to help readers evaluate both measures in our study.

In conclusion, our study found that the shortened version of the C-BARQ is a valid tool to evaluate the behavior of dogs by their owners and a complement to the long C-BARQ.

## Supporting information

**S1 Table.**
(CSV)

## Acknowledgments

We would like to thank Dr. James Serpell and the University of Pennsylvania for allowing us to use both the shortened C-BARQ as well as the full-length questionnaire for this project. The authors also thank Dog Aging Project participants, their dogs, and community veterinarians for their important contributions.

Full list of Dog Aging Project Consortium (consortium@dogagingproject.org) members and their affiliations (as of January 1, 2024): Joshua M. Akey[1], Brooke Benton[2], Elhanan Borenstein[3,4,5], Marta G. Castelhano[6], Amanda E. Coleman[7], Kate E. Creevy[8], Kyle Crowder[9,10], Matthew D. Dunbar[10], Virginia R. Fajt[11], Annette L. Fitzpatrick[12,13,14], Unity Jefrey[15], Erica C. Jonlin[2,16], Elinor K. Karlsson[17,18], Jonathan M. Levine[8], Jing Ma[19], Robyn L. McClelland[20], Daniel E.L. Promislow[2,21], Audrey Ruple[22], Stephen M. Schwartz[13,23], Sandi Shrager[24], Noah Snyder-Mackler[25,26,27], Silvan R. Urfer[2], Benjamin S. Wilfond[28,29]

[1]Lewis-Sigler Institute for Integrative Genomics, Princeton University, Princeton, NJ, USA
[2]Department of Laboratory Medicine and Pathology, University of Washington School of Medicine, Seattle, WA, USA

[3]Department of Clinical Microbiology and Immunology, Sackler Faculty of Medicine, Tel Aviv University, Tel Aviv, Israel

[4]Blavatnik School of Computer Science, Tel Aviv Univer sity, Tel Aviv, Israel

[5]Santa Fe Institute, Santa Fe, NM, USA

[6]Cornell Veterinary Biobank, College of Veterinary Medi cine, Cornell University, Ithaca, NY, USA

[7]Department of Small Animal Medicine and Surgery, College of Veterinary Medicine, University of Georgia, Athens, GA, USA

[8]Department of Small Animal Clinical Sciences, Texas A&M University College of Veterinary Medicine & Biomedical Sciences, College Station, TX, USA

[9]Department of Sociology, University of Washington, Seattle, WA, USA

[10]Center for Studies in Demography and Ecology, University of Washington, Seattle, WA, USA

[11]Department of Veterinary Physiology and Pharmacology, Texas A&M University College of Veterinary Medicine & Bio medical Sciences, College Station, TX, USA

[12]Department of Family Medicine, University of Washing ton, Seattle, WA, USA

[13]Department of Epidemiology, University of Washington, Seattle, WA, USA

[14]Department of Global Health, University of Washington, Seattle, WA, USA

[15]Department of Veterinary Pathobiology, Texas A&M University College of Veterinary Medicine & Biomedical Sciences, College Station, TX, USA

[16]Institute for Stem Cell and Regenerative Medicine, University of Washington, Seattle, WA, USA

[17]Bioinformatics and Integrative Biology, University of Massachusetts Chan Medical School, Worcester, MA, USA

[18]Broad Institute of MIT and Harvard, Cambridge, MA, USA

[19]Division of Public Health Sciences, Fred Hutchinson Cancer Research Center, Seattle, WA, USA

[20]Department of Biostatistics, University of Washington, Seattle, WA, USA

[21]Department of Biology, University of Washington, Seattle, WA, USA

[22]Department of Population Health Sciences, Virginia-Maryland College of Veterinary Medicine, Virginia Tech, Blacks burg, VA, USA

[23]Epidemiology Program, Fred Hutchinson Cancer Research Center, Seattle, WA, USA

[24]Department of Biostatistics, Collaborative Health Studies Coordinating Center, University of Washington, Seattle, WA, USA

[25]School of Life Sciences, Arizona State University, Tempe, AZ, USA

[26]Center for Evolution and Medicine, Arizona State University, Tempe, AZ, USA

[27]School for Human Evolution and Social Change, Arizona State University, Tempe, AZ, USA

[28]Treuman Katz Center for Pediatric Bioethics, Seattle Children's Research Institute, Seattle, WA, USA

[29]Department of Pediatrics, Division of Bioethics and Palliative Care, University of Washington School of Medicine, Seattle, WA, USA

## Author Contributions

**Conceptualization:** Jeremy Evans, Annette L. Fitzpatrick, Kate E. Creevy, Audrey Ruple.

**Data curation:** Jeremy Evans, Audrey Ruple.

**Formal analysis:** Vanessa Wilkins, Christina Park, Annette L. Fitzpatrick, Audrey Ruple.

**Methodology:** Annette L. Fitzpatrick, Audrey Ruple.

**Supervision:** Annette L. Fitzpatrick, Kate E. Creevy, Audrey Ruple.

**Writing – original draft:** Vanessa Wilkins, Jeremy Evans, Christina Park, Annette L. Fitzpatrick, Kate E. Creevy, Audrey Ruple.

**Writing – review & editing:** Vanessa Wilkins, Christina Park, Annette L. Fitzpatrick, Kate E. Creevy, Audrey Ruple.

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
