## [Decision Letter · Decision Letter 0]

16 Nov 2023

PONE-D-23-25264Validation of the shortened version of the Canine Behavioral Assessment and Research Questionnaire (C-BARQ) using participants from the Dog Aging ProjectPLOS ONE Dear Dr. Ruple,

Thank you for submitting your manuscript to PLOS ONE. After careful consideration, we feel that it has merit but does not fully meet PLOS ONE’s publication criteria as it currently stands. Therefore, we invite you to submit a revised version of the manuscript that addresses the points raised during the review process.

**ACADEMIC EDITOR: **

REVIEWER #1:

This work is very interesting and also very hard as answering to surveys require a lot of time and effort from the owners, and I admire this type of work very much.

A few questions arose while going over the work, which I will need some clarity on:

Do you have the demographics of the owners who filled out the questionnaire? Such as gender, age, level of education and social class?

Do you believe the gender or other characteristics of the owners who responded to the questionnaire could influence their responses? On that same note, I believe if table 1 contained demographic information about the owners, it could reduce possible biases and/or raise new questions.

Do you know if the same owners responded to both the long and short questionnaires and if they responded at the same time? I raise this question because the questionnaires are long which could influence the effort the owners put into responding the question towards the end of the survey.

On the discussion, I felt the point of the possible reason on aggressiveness was missed. Could it be because the questions are ambiguous and hard to interpret which can be answered in different ways? I also believe it’s relevant to discuss how answers regarding gender and size/type of breed can influence on the animal’s aggressiveness? Is there any type of association already described?

Reviewer #2

I think the analysis was performed rigorously and explained in details. I am concerned about the 60% response rate. While the authors did a great job of mentioning it in the limitations, I think more information can be included about why they think this happened, how it may impact the results, and how the authors are going to mitigate the effects.

We look forward to receiving your revised manuscript.

Kind regards,

Colleen Anne Dell, Ph.D.

Academic Editor

PLOS ONE

Journal Requirements:

2. You indicated that ethical approval was not necessary for your study. We understand that the framework for ethical oversight requirements for studies of this type may differ depending on the setting and we would appreciate some further clarification regarding your research. Could you please provide further details on why your study is exempt from the need for approval and confirmation from your institutional review board or research ethics committee (e.g., in the form of a letter or email correspondence) that ethics review was not necessary for this study? Please include a copy of the correspondence as an ""Other"" file.

7. One of the noted authors is a group or consortium [The Dog Aging Project Consortium]. In addition to naming the author group, please list the individual authors and affiliations within this group in the acknowledgments section of your manuscript. Please also indicate clearly a lead author for this group along with a contact email address.

8. Please include your full ethics statement in the ‘Methods’ section of your manuscript file. In your statement, please include the full name of the IRB or ethics committee who approved or waived your study, as well as whether or not you obtained informed written or verbal consent. If consent was waived for your study, please include this information in your statement as well. 

Additional Editor Comments:

Dear Authors,

The external review on your paper has been completed but two independent academics. I am happy to inform you that your paper is being accepted with minor revisions.

Please attend to the specific comments below:

REVIEWER #1:

This work is very interesting and also very hard as answering to surveys require a lot of time and effort from the owners, and I admire this type of work very much.

A few questions arose while going over the work, which I will need some clarity on:

Do you have the demographics of the owners who filled out the questionnaire? Such as gender,

age, level of education and social class?

Do you believe the gender or other characteristics of the owners who responded to the

questionnaire could influence their responses? On that same note, I believe if table 1 contained

demographic information about the owners, it could reduce possible biases and/or raise new

questions.

Do you know if the same owners responded to both the long and short questionnaires and if

they responded at the same time? I raise this question because the questionnaires are long

which could influence the effort the owners put into responding the question towards the end

of the survey.

On the discussion, I felt the point of the possible reason on aggressiveness was missed. Could it

be because the questions are ambiguous and hard to interpret which can be answered in

different ways? I also believe it’s relevant to discuss how answers regarding gender and size/type

of breed can influence on the animal’s aggressiveness? Is there any type of association already

described?

REVIEWER #2:

I think the analysis was performed rigorously and explained in details. I am concerned about the 60% response rate. While the authors did a great job of mentioning it in the limitations, I think more information can be included about why they think this happened, how it may impact the results, and how the authors are going to mitigate the effects.

Please also review the article generally for flow, spelling, grammar, etc. now that it has been a few weeks since it was originally submitted.

Thank you,

Colleen

Reviewers' comments:

Reviewer's Responses to Questions

**Comments to the Author**

1. Is the manuscript technically sound, and do the data support the conclusions?

Reviewer #1: Yes

Reviewer #2: Yes

2. Has the statistical analysis been performed appropriately and rigorously? 

Reviewer #1: Yes

Reviewer #2: Yes

3. Have the authors made all data underlying the findings in their manuscript fully available?

Reviewer #1: Yes

Reviewer #2: Yes

4. Is the manuscript presented in an intelligible fashion and written in standard English?

Reviewer #1: Yes

Reviewer #2: Yes

5. Review Comments to the Author

Reviewer #1: This work is very interesting and also very hard as answering questionnaire require a lot of time and effort from the owners, and I admire this type of work very much.

I don´t have any questions about method and statistical analysis, and no other concerns regarding the publication. I think the kappa statistic was a good choice. I just suggest a more robust discussion as detailed comments on the attached file.

Reviewer #2: I think the analysis was performed rigorously and explained in details. I am concerned about the 60% response rate. While the authors did a great job of mentioning it in the limitations, I think more information can be included about why they think this happened, how it may impact the results, and how the authors are going to mitigate the effects.

6. PLOS authors have the option to publish the peer review history of their article (what does this mean?). If published, this will include your full peer review and any attached files.

Reviewer #1: **Yes: **Mariana Yukari Hayasaki Porsani

Reviewer #2: No

---

## [Author Response · Author response to Decision Letter 0]

9 Feb 2024

Thank you both for your reviews and comments. Our specific responses follow below.

REVIEWER #1:

This work is very interesting and also very hard as answering to surveys require a lot of time and effort from the owners, and I admire this type of work very much.

Thank you so much for this comment. We are excited to be able to present some of the work we are doing with data collected at the Dog Aging Project.

A few questions arose while going over the work, which I will need some clarity on:

Do you have the demographics of the owners who filled out the questionnaire? Such as gender,

age, level of education and social class?

In order to adhere to our IRB exemption, we only collect minimal information about the dog owners. That information includes age categories, highest level of education, and annual income categories. We do not collect any information about sex or gender and owners are able to select “prefer not to answer” for the income question.

Do you believe the gender or other characteristics of the owners who responded to the

questionnaire could influence their responses? On that same note, I believe if table 1 contained

demographic information about the owners, it could reduce possible biases and/or raise new

questions.

We really appreciate this suggestion. We have added owner demographic information (age, education level, and income) to table 1. Due to the limitations regarding human-level data collection mentioned above, the owner information is categorized and thus would likely limit our ability to infer a great deal from it. However, this could be hypothesis generating for future studies.

Do you know if the same owners responded to both the long and short questionnaires and if

they responded at the same time? I raise this question because the questionnaires are long

which could influence the effort the owners put into responding the question towards the end

of the survey.

Only owners who completed both versions of the survey were included in this analysis. The short version of the survey was completed first in July of 2020 as part of the HLES survey. The long version of the survey was completed in August of 2020 as a standalone survey. Thus, even though the survey itself was longer, there were no other surveys being completed at the same time. Information about the timing of the survey completions is included in the methods subsection entitled “Study population.”

On the discussion, I felt the point of the possible reason on aggressiveness was missed. Could it

be because the questions are ambiguous and hard to interpret which can be answered in

different ways? I also believe it’s relevant to discuss how answers regarding gender and size/type

of breed can influence on the animal’s aggressiveness? Is there any type of association already

described?

We limited the discussion about the specific results of the surveys as our main purpose was to validate that similar responses can be obtained from owners of dogs with multiple temperament types using either version of the C-BARQ survey (long or short). We felt that an in depth discussion about the causes of aggressiveness in dogs – or really causes of why owners report aggressiveness in dogs – would detract from that purpose. We do agree that there can be misinterpretation of questions, but feel that we have shown that regardless of the accuracy, owners report similar responses using both survey types.

REVIEWER #2:

I think the analysis was performed rigorously and explained in details. I am concerned about the 60% response rate. While the authors did a great job of mentioning it in the limitations, I think more information can be included about why they think this happened, how it may impact the results, and how the authors are going to mitigate the effects.

Thank you so much for taking the time to review this manuscript. We appreciate your effort and comments. We feel that the 60% response rate at the height of the pandemic and pre-vaccine was acceptable, but agree that we needed to increase the discussion about that point. The discussion now reads: “Second, while the response rate at nearly 60% was acceptable, particularly given the timing of the survey request in relation to world events, the potential for non-response bias is worth mentioning. For instance, it could be that participants that were not able to complete the long version of the C-BARQ survey were those who were no longer under quarantine restrictions in place due to the pandemic and thus their dogs may have been experiencing a shift in their daily routine that could have led to less agreement between survey responses than within the population who was able to complete both surveys.”

Please also review the article generally for flow, spelling, grammar, etc. now that it has been a few weeks since it was originally submitted.

Thank you – we hope that it reads smoothly, but are happy to adjust as needed!

---

## [Editor Report · Decision Letter 1]

20 Feb 2024

Validation of the shortened version of the Canine Behavioral Assessment and Research Questionnaire (C-BARQ) using participants from the Dog Aging Project

PONE-D-23-25264R1

Dear Dr. Audrey Ruple,

We’re pleased to inform you that your manuscript has been judged scientifically suitable for publication and will be formally accepted for publication once it meets all outstanding technical requirements.

Kind regards,

Colleen Anne Dell, Ph.D.

Academic Editor

PLOS ONE
---

## [Editor Report · Acceptance letter]

2 Apr 2024

PONE-D-23-25264R1 

PLOS ONE

Dear Dr. Ruple, 

I'm pleased to inform you that your manuscript has been deemed suitable for publication in PLOS ONE. Congratulations! Your manuscript is now being handed over to our production team.

Kind regards, 

on behalf of

Dr. Colleen Anne Dell 

Academic Editor

PLOS ONE